# Use of Neural Networks for Lifetime Analysis of Teeming Ladles

**DOI:** 10.3390/ma15228234

**Published:** 2022-11-19

**Authors:** Dalibor Jančar, Mario Machů, Marek Velička, Petr Tvardek, Leoš Kocián, Jozef Vlček

**Affiliations:** 1Department of Thermal Engineering, Faculty of Materials Science and Technology, Institute of Environmental Technology, VSB-Technical University of Ostrava, 17. Listopadu 2172/15, 708 00 Ostrava, Czech Republic; 2Liberty Ostrava a.s., Vratimovská 689/117, 719 00 Ostrava, Czech Republic

**Keywords:** neural networks, refractory material, ladle, modelling

## Abstract

When describing the behaviour and modelling of real systems, which are characterized by considerable complexity, great difficulty, and often the impossibility of their formal mathematical description, and whose operational monitoring and measurement are difficult, conventional analytical–statistical models run into the limits of their use. The application of these models leads to necessary simplifications, which cause insufficient adequacy of the resulting mathematical description. In such cases, it is appropriate for modelling to use the methods brought by a new scientific discipline—artificial intelligence. Artificial intelligence provides very promising tools for describing and controlling complex systems. The method of neural networks was chosen for the analysis of the lifetime of the teeming ladle. Artificial neural networks are mathematical models that approximate non-linear functions of an arbitrary waveform. The advantage of neural networks is their ability to generalize the dependencies between individual quantities by learning the presented patterns. This property of a neural network is referred to as generalization. Their use is suitable for processing complex problems where the dependencies between individual quantities are not exactly known.

## 1. Introduction

One of the most critical aggregates of a steel plant is the teeming ladle. The lining, made of high-quality, heat-resistant materials, guarantees its long service life. Choosing the appropriate refractory lining requires knowledge of the circulation of the casting ladle during production and knowledge of the individual influences aggressively acting on the refractory lining. As a result of the multiple uses of the ladle, its lining wears out during the ladle campaign. This wear is caused by the erosive and corrosive effect of molten metal and slag. Slowing down the wear process of the lining can be guaranteed not only by choosing the type of refractory materials but also by other measures, especially by a suitable thermal regime of the ladle. Maintaining the temperature of the lining of the casting ladle at high, and, if possible, constant temperatures positively affects the degree of wear of the lining [1]. It is evident that many operating factors affect the lining of the ladle and that there is a mutual connection between them, i.e., that the chosen lifetime solution method must be able to compare a large set of non-linear data [2].

Digitalization of production technologies is also essential in energy intensive industries, such as the steel industry, where emphasis is placed on optimizing the production chain and sustainable production [3]. Artificial intelligence tools [4] are used to estimate the ageing of ladle furnaces, where real data are used for tuning the model and then identifying less critical situations at the end of the ladle’s life [5]. Another possibility is to establish a mechanism for predicting the service life of composite ladle structures, which is based on the stress analysis of the steel shell of the ladle and combines conventional fatigue analysis with extended fracture theory. These techniques allow the prediction of the service life of the steel shell by detecting its crack length [6]. A mathematical model of heat transfer through the lining based on technological characteristics and thermal imaging images can be used to diagnose the thickness of the casting ladle in critical areas [7]. Different diagnostics obtain directly measured data using a thermal imaging camera, allowing the determination of the residual thickness of the refractory lining of the metallurgical ladle [8]. For predicting refractory wear of furnaces, linear models are used, which work on the principle of the Kalman filter [9]. For processing a large volume of data intended for diagnostics, the Bayesian network model can be used, which uses the Hadoop software structure and thus achieves a high efficiency of knowledge reasoning [10].

Moreover, special models are proposed for diagnostics and predictions of the current condition of the lining in high-temperature objects in metallurgy [11]. LabVIEW software can also create and monitor the casting ladle system database, which collects data on ladle usage and filling [12]. The use of neural networks has seen a significant upward trend in recent years [13,14]. With the help of neural networks, the need for maintenance can be predicted based on an analysis of the time series of the main parameters of the production equipment [15]. The application of neural networks for device diagnostics is extensive; it is used, for example, to detect critical zones in basins used for the marine industry [16]. Neural networks are also used for the metallographic quality control of metals and for the recognition of their microstructures [17].

The lifetime assessment of casting ladles was carried out using the method of neural networks. A set of input data (operating factors affecting the lifetime of the lining) was compared with an output parameter (number of castings, i.e., the number of cycles on a continuous casting machine, the so-called CMM).

## 2. Materials and Methods

In general, every computer can be understood as a device that displays a set of input data (e.g., coefficients of a linear system) on a set of other data (e.g., the solution of this system). Neural networks belong to the machine learning methods together with linear regression, decision tree models, support vector machines, probabilistic modelling, or genetic algorithms [18]. If we do not impose further simplifications, it can be stated that even the nervous system of an individual animal is a kind of prototype (every one is different) of a computer. Much mental activity or vegetative activity of an animal can be considered as the implementation of some algorithm (e.g., a more or less meaningful answer to a more or less meaningful question, a reaction to a given environmental stimulus, etc.) [19]. However, there are fundamental differences between these two parallel worlds of living computers and classical electronic computers. Let us try to summarize those that we believe are related to the principle of their operation (see Table 1):

It can be expected that an analysis of the structures, which comprise a straightforward model of the nerve centres of animals (based on the currently known findings from the fields of anatomy, biochemistry, and neurology), can be beneficial both for the research of living organisms and vice versa. The development and complementation of new types of parallel computer architectures will make it possible to solve problems for which computers have no effective algorithm [20].

Artificial neural networks find applications in damaged pattern reconstruction, classification, database searching, prediction, approximation, extrapolation, image recognition, and other fields. They can resist both their damage and errors in the input data. Its advantage is a parallel structure allowing for increased speed. They are vital when solving tasks whose data structure contains non-linearities [21].

### 2.1. Artificial Neuron

An artificial neuron is a simplified mathematical representation of the function of a biological neuron. A formal neuron (see Figure 1) has n generally real inputs *x*_1_, …, *x*_n_ corresponding to dendrites. All inputs are evaluated by the respective synaptic weights *w*_1_, …, *w*_n_, which are generally also real.

The weights determine the degree of the throughput of the input signal. The weighted sum of the input values represents the internal potential of the neuron *z*:(1)$$z=\overset{n}{\underset{i=1}{\sum }}w_{i}\cdot x_{i}-h$$
where *h* is the threshold value of the neuron (1), *w* is the synaptic weight (1), *x* is the input value (1), *z* is the internal potential of the neuron (1).

The output (state) of the neuron *y* modelling the electrical impulse of the axon is given by a generally non-linear transfer function whose argument is the internal potential *z*. This output is then, at the same time, the input given to other neurons, which is shown in Figure 2.

Among the most important mathematical models consisting of one, a neuron has a continuous perceptron whose potential is defined as a weighted sum of incoming signals. Its transfer function is continuous and differentiable. The most commonly used transfer functions include:hyperbolic tangents;sharp non-linearity (jump function);linear function;standard (logistic) sigmoid;hyperbolic tangent.

Historically, the first unit step function (Heaviside step function) was used (Figure 3). It follows from the course of this function and the principle of network operation that, with the exclusive use of this transfer function in the entire network, it is possible to only request a two-state neuron output, which can be disadvantageous for a number of practical applications.

Another possible type of transfer function is a linear or piecewise linear function (Figure 4). Research has shown that, when these functions are used, the quality of the network in terms of adaptation speed and generalization ability is lower than that of networks with a non-linear transfer function.

The presence of linear transfer functions entails one disadvantage: these functions are not resistant to disturbances, i.e., if an excessively large value comes to the input of the neuron, then the argument is transferred to the output in proportion to the direction of the linear function. However, their use is particularly important for the output layer of the neural network when it is not necessary to transform the required quantities into the interval 0-1, as is often necessary with other transfer functions that move in this interval [22,23].

The third and, at the same time, the most used type of functions are non-linear continuous monotone functions. This group’s two most used functions are the sigmoid (Figure 5) and the hyperbolic tangent (Figure 6). Such transfer functions allow the neuron’s sensitivity to be significant for small signals. In contrast, for larger signal levels, its sensitivity decreases. They are, therefore, highly resistant to distortions. If an excessively large value arrives at the neuron’s input due to the curvature of these functions, values will appear at the output either close to 1 or 0 (−1), depending on the sign of the argument. The steepness of the sigmoid can be taken as a parameter of the network, which contributes significantly to the quality of its activity. Steepness also affects the learning (or adaptation) speed of the network. Therefore, algorithms were developed that, based on the dynamics of the decline of the so-called error function of the network, adjust the values of the steepness of the transfer functions of the entire network or individual neurons [24,25].

The equation of the standard sigmoid (Figure 5) has the form:(2)$$y=\sigma \left(z\right)=\frac{1}{1+e^{-\lambda_{s}z}}$$
where *λ*_s_ indicates the slope of the sigmoid and *y* represents the excitation value.

The tangent equation (Figure 6) has the form:(3)y=tanh(z)
where the internal potential *z* is given by Equation (1).

The following conclusions follow from the above facts:the excitation of the neuron varies between 0 and 1, where the value 1 means full excitation of the neuron in contrast to the value 0, which corresponds to the state of inhibition;if the internal potential of the neuron approaches the value +∞, then the neuron is fully excited, i.e., *y* = 1;on the contrary, if the internal potential of the neuron approaches the value −∞, the neuron is completely inhibited, i.e., *y* = 0.

### 2.2. Neural Network

Neural networks can have one, two, or three layers, or even thousands of them, in case of deep-learning tasks. More layers increase the time required for learning. There can be multiple outputs from the neural network, or there can be only one output. In multi-layer networks, the first layer is always branching, which means that the neurons in the input layer only distribute the input values to the next layer. Since it is generally a multi-point entry into the network, we are talking about entry or output vectors of information. The required number of neurons in individual layers is variable and depends on the problem being solved. A much more appropriate way to determine the number of neurons is to use a network that itself changes this number according to the evolution of the global error [22]. Each input to the neuron is assigned a so-called weight *w*. Weight *w* is a dimensionless number that determines how significant a given input is to the respective neuron (not the network or problem). The learning ability of neural networks lies precisely in the ability to change all the weights in the network according to suitable algorithms—in contrast to biological networks, which are based on the ability to create new connections between neurons.

Physically, both learning abilities are based on different principles, but not logically. In the case of the creation of a new connection (input) in a biological neuron, it is the same as when, in a technical network, the connection between two neurons is initially evaluated with a weight of 0 (and therefore does not exist as an input for the neuron into which it enters) and at the moment when the weight changes to a non-zero number and the given connection becomes visible, i.e., when it is created [26]. Nowadays, there are many types of neural networks, each of which has a predetermined use according to its structure. For most possible applications, the feed-forward, multi-layer neural network is still the most preferred neural network [25].

The most well-known type of feed-forward neural network is the Perceptron. A schematic of a simple perceptron is shown in Figure 7. Essential characteristics:simple neuron, or single layer forward propagation network;learning: with a teacher;neuronal activation function: signum function, or unit jump.

**Figure 7 materials-15-08234-f007:**
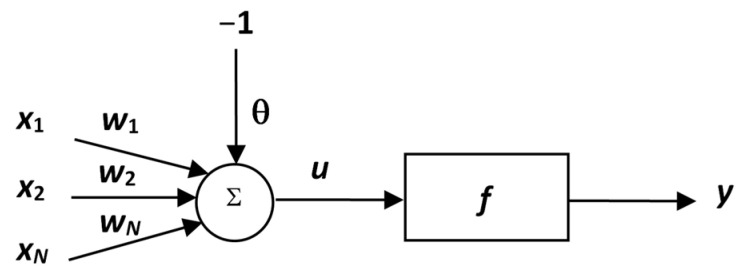
Simple perceptron.

Utilization:classification of linearly separable data (see Figure 8).

The best-known structure of neural networks used today is the multi-layer perceptron (MLP), which operates with a linear function and, by default, with a non-linear sigmoid activation function [27]. Many examples of MLP use in predictive maintenance, failure analysis, or defect occurrence prediction in industrial processes can be found in the literature, e.g., in works [28,29,30,31].

In some cases, a hyperbolic function (tanh) is also used, producing better results. MLP can be used both for quasi-regression tasks and for classification tasks. With different types of features and customizations, it can serve well for both types of tasks. For classification, MLP is used for simple decision-making (single output) and multiple classifications (multi-output). MLP networks are compact with a wide range of applications. MLP training algorithms include Back Propagation, Quick Propagation, Levenberg–Marquardt, Delta-bar-Delta, and several other modifications of training algorithms. Its significant advantage is its very fast solving of tasks. The disadvantage is slow training. An illustration of an MLP network with four layers is shown in Figure 9 [32].

### 2.3. Learning Multi-Layer Feed-Forward Neural Networks

Multi-layer, feed-forward neural networks are among the most widely used neural network models. The fundamental element of this type of network is a continuous perceptron. The network consists of at least three layers of neurons: input, output, and at least one inner layer, or hidden layer. There is always a so-called complete connection of neurons between two adjacent layers, i.e., each neuron of a lower layer is connected to all neurons of a higher layer. There are no cycles or mutual connections of neurons from the same layer.

The input to this type of network is a vector of numbers, as many as there are neurons in the input layer, since each input layer neuron has only one input. The neurons in the input layer do not perform any mathematical operations with the input values. The only task of these neurons is to distribute the input signals to the next layer so that each neuron of the first hidden layer has the entire input vector at its inputs. Signal propagation in this type of network is feed-forward propagation.

Learning aims to determine the transformation function using a particular criterion. This error function expresses the general relationship between the input and output variables with a certain approximation. Finding such a relationship means solving a specific problem area. The network learning consists of presenting individual patterns (vectors) from the training set, i.e., the individual elements forming ordered input–output pairs. Each pattern of the training set describes how the input and output layer neurons are excited.

We obtain the network’s response by presenting an input vector and known network signal propagation. It will initially be quite different from the output of the specified training set. Learning means adapting the weights so that the network’s responses differ as little as possible from the desired outputs in the training set [33].

We can formally define the training set *T* as a set of elements (patterns) that are defined as ordered pairs in the following way:(4)$$\begin{array}{c} T=\left\{\left[I_{1},O_{1}\right],\left[I_{2},O_{2}\right]\ldots \left[I_{p},O_{p}\right]\right\}\\ I_{i}=\left[i_{1},i_{2}\ldots i_{m}\right]i_{j}=\langle 0,1\rangle \\ O_{i}=\left[o_{1},o_{2}\ldots o_{n}\right]o_{j}=\langle 0,1\rangle \end{array}$$
where *p* is the number of training patterns, *I_i_* is the vector of excitations of the input layer, which is made up of *m* neurons, *O_i_* is the vector of excitations of the output layer (required responses to *I_i_*), which is made up of *n* neurons, *i_j_*, *o_j_*—is the excitation of the *j*th input neuron and output layers.

## 3. Results

For our solution, data from the 230-ton casting ladle of Liberty Ostrava a.s. steelwork were used. The ladle lining is shown in Figure 10. The wall and bottom of the working lining is made of AMC fittings with a composition of 84 % Al_2_O_3_; 8.5% MgO; 7% residual carbon; 2.5% SiO_2_. The lining in the area of the MgO-C slag line has a composition of 95% MgO and 10% residual carbon.

In order to collect data, i.e., input data, application software was created, in which data from individual database structures created in the control system of the steel mill are processed. Following is a complete data set containing data:from the circulation subsystem of the ladles;from the so-called furnace side of the steelworks (tandem open hearth furnace, steel processing in a ladle furnace);from the so-called displacement side (continuous casting of steel).

The data in this application software is designed to map a ladle campaign from laying to decommissioning. Data from individual ladle cycles are always included within the given campaign. Obtained data analysis and subsequent data cleaning were carried out using the functions of the Visual Basic for Applications (VBA) programming language. The resulting file, containing 106 ladle campaigns, was subsequently used to evaluate the life of the ladle lining. It should be noted here that different steel grades were processed in the campaigns used, and therefore the obtained results are applicable to any steel company.

The problem of the influence of operational factors on the life of the lining of the casting ladle was solved in the first phase using multiple regression analysis in MS Excel. The so-called Student’s *t*-test was performed to investigate the statistical significance of the individual parameters on the output parameter (number of castings). Based on this, some parameters with very little significance for the output parameter were discarded from the tested set. In the second phase, the solution was implemented in the Gensym product environment, the so-called NeurOn-Line Studio (Gensym Corporation, Austin, Texas, USA). This is a graphical, object-oriented software product for creating neural network applications. Using NeurOn-Line Studio, it is possible to create dynamic and non-linear models without knowing the structure or analytical description of the model. Data stored in databases, historians or text files, are sufficient to create a model.

### 3.1. Learning of Operational Factors

Among the many operating factors that were predicted to have the most significant effect on ladle lining life, the following were selected:empty ladle time (end of pouring at continuous casting machine (CCM) until tapping)—this is the time from the end of pouring to the next tapping when the ladle is empty. This period usually includes high-temperature heating of the ladle lining;full ladle period—this is the period from tapping to the end of casting, when the ladle is full of steel, or the steel level drops during casting;tapping temperature—this is the temperature of the steel when tapped from the given melting unit. In our case, it was a tandem furnace;steel temperature in the ladle after tapping—since the range of the recommended temperature of steel in the ladle after tapping of the brand of steel produced is set in the technological regulations of the steel plant, this value is not always the same for a certain brand; however, this temperature varies in a certain interval;electricity consumption (when heating steel in a ladle furnace)—the effect of heating steel in a ladle furnace on the wear of the ladle lining is represented in our case by the total electricity consumption for melting;the number of melts processed in the caisson of the vacuum station—the effect on the lining can be expressed by the number of ladle cycles when a vacuum station is included in the production process;argon consumption—for this evaluation, the total argon consumption per melt was used.

### 3.2. Testing the Training Model

The procedure for creating a model in NeurOn-Line Studio is as follows:

1. The data are imported first. NeurOn-Line Studio uses data in two basic types. The first type is time-based data, so-called Time-Based Data, where a timestamp is attached to each record. The second type is time-independent data, so-called Row-Based Data. Data import loading data from a text file is similar to loading data from a text file in MS Excel. The imported data can be processed further using the built-in editor and their visualization in graphs;

2. Creation of the so-called preprocessor allows us to select data groups suitable for training;

3. After determining the input variables to the neural networks and the output values to be learned by the model, training or learning of the neural networks follows. The model creator or user can monitor the entire course of training. The training of the network is shown in Figure 11. The left part of the figure represents the course of the least squares deviation between the training results and the actual values, and the right graph shows the correlation between the predicted and the real value of the output.

The user is notified of the just-completed training by a message in the middle of the screen. The result is graphically represented by the relationship between the predicted and actual values of the output parameter of the model (Figure 12). Figure 11 and Figure 12 show the training in the NeurOn-Line Studio program. They are not the resulting graphs for the selected periods [35].

The result of the training is a prediction model that can be used using the ActiveX component in any common programming environment under MS Windows or is directly usable in the G2 environment with the NeurOn-Line superstructure. For the model, it is possible to view the basic statistics of the Root Mean Squared Error (RMSE) model and the correlation coefficient immediately after training. The model can then be analyzed using standard statistical methods [36].

### 3.3. Using the Model for Real Conditions

The actual evaluation of the analysis results was carried out by comparing the set of input data mentioned above and the output parameter (the number of castings). Figure 13 shows the output after training the set, where n and the x-axis are the actual numbers of the castings and the y-axis is the number of simulated (predicted) castings.

Therefore, the graph describes to what extent the neural network was able to learn from the given data. The exact correlation coefficient can also be read in the NeurOn-Line Studio environment. In simple terms, it can be said that, if the red line leads from the lower left corner to the upper right corner, it is a very well-trained neural network, i.e., a network with a high correlation coefficient. In this case, the correlation coefficient R^2^ was 0.7938.

On the trained data set, simulations of the influence of selected operational factors on lining wear were carried out for both periods while maintaining the interrelationships of other parameters. This analysis was carried out to determine how a change in a particular technological parameter would affect the durability of the ladle lining under the given conditions of steel production and processing, i.e., taking into account the simultaneous action of all factors occurring during ladle cycling. Figure 14 shows the dependencies of four selected parameters affecting lining wear. In the graphs, the x-axis expresses the given parameter and the y-axis the number of castings. At the point where the x-axis crosses the y-axis, on the x-axis, the minimum value of the given parameter is selected either from the input values for the neural network or the minimum possible, which is allowed by the current state of the technological process of steel production in the steel mill.

Similarly, the maximum value is left as the maximum of the input set of the given parameter, or an extreme state that could occur in a real state was chosen. It is clear from the graphs that, as the time of the empty ladle increases and the amount of argon blown into the ladle increases, so too does the number of castings, which indicates a positive effect of these factors on the life of the lining. The other two factors, i.e., electricity consumption and tap temperature, have the exact opposite effect.

The result of the simulation expressing the effects of all selected operating parameters on the life of the lining is shown in a bar graph in Figure 15.

The bar chart describes the percentage representation of individual parameters, either with positive influence (plus sign in the column) or negative (minus sign in the column). The biggest negative influence is the time the ladle is empty, and the consumption of electricity and subsequently is the number of vacuuming cycles.

The very negative effect of the empty ladle time can be explained by temperature shocks in the lining during its constant cooling and reheating. These large temperature changes can change the coefficient of thermal expansion in the surface layer with respect to the expansion of the original material, which, even with slight temperature changes, induces shear stress at the interface of the layers sufficient for the formation of cracks. This structural spalling progresses layer by layer and can lead to rapid wear and tear of the lining. The best functioning ladle economy occurs if the lining is kept at high and above all constant temperatures, which requires continuous cycling of the ladle, without the intermediate heating associated with covering the ladle with a lid during transport. Two other factors—the consumption of electrical energy and the number of vacuuming cycles—increase the temperature of the steel in the pan, which accelerates the degradation effects of refractory materials, especially its corrosion. Again, both of these effects could be reduced by cycling the ladle, thereby preserving the overall enthalpy of the lining. The consumption of argon blown into the steel through the bottom plug of the ladle has a positive effect on the life of the lining. We believe this is due to the perfect temperature homogenization of the steel, which causes equalization of temperatures in the lining of the entire ladle. The total amount of blown argon enters analysis as a single constant value for particular heat without differentiation between the blowing phases.

## 4. Conclusions

As part of the statistical evaluation of the service life of the linings of the casting ladles in the steel plant, application software was created in which data from individual database structures of the control system of the steel plant are processed. The data in this application software is designed to map a ladle campaign from laying to decommissioning. This means that data from individual ladle cycles are always included within the given campaign. These data were subsequently evaluated in neural networks in the Gensym product environment, the so-called NeurOn-Line Studio, a graphical, object-oriented software product for creating neural network applications. For this analysis, a set of input data was created and compared with the output factor, which is the number of castings of the given ladle in the campaign, i.e., the number of cycles of the continuous casting machine, the so-called CCM. The resulting value is the percentage representation of the individual tested parameters on the life of the lining of the casting ladle, which is shown in Figure 15. The most significant adverse effect is the time of the ladle’s emptiness, electricity consumption, and, subsequently, the number of vacuuming cycles; moreover, argon blown into the ladle had a very positive effect.

To increase the life of the ladle lining, therefore, it is recommended to cycle it as quickly as possible, if possible, without including high-temperature heating between melts. If the ladle has a sufficiently high enthalpy of the lining before tapping, reduce the time between the end of tapping and the ladle furnace to a minimum, limiting heating by the electrodes on the ladle furnace. The ladle must be covered with a lid at every possible moment to preserve the enthalpy accumulated in the masonry of the lining.

The main innovation of the research lies in the use of neural networks in the field of basin metallurgy. The neural network method has been proven to be a suitable tool for solving the lifetime of casting ladles, especially when compared to classical multiple regression methods. Based on this solution, measures were designed to ensure an increase in the life of the linings of the casting ladles.

Currently, further data collection is taking place, which will expand research in this area. The authors intend to use machine intelligence algorithms to refine and verify the results achieved. The obtained results will be used in the field of predicting the lifetime of the lining of casting ladles in the real operation of a steel plant. The authors also want to use the latest methods of neural networks in other areas of the metallurgical industry.

## Figures and Tables

**Figure 1 materials-15-08234-f001:**
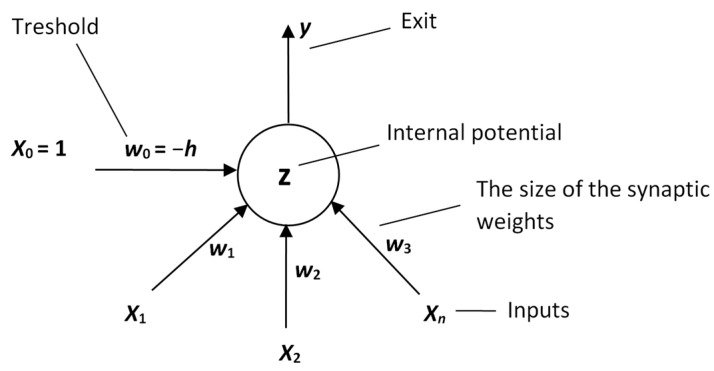
Formal neuron schema.

**Figure 2 materials-15-08234-f002:**
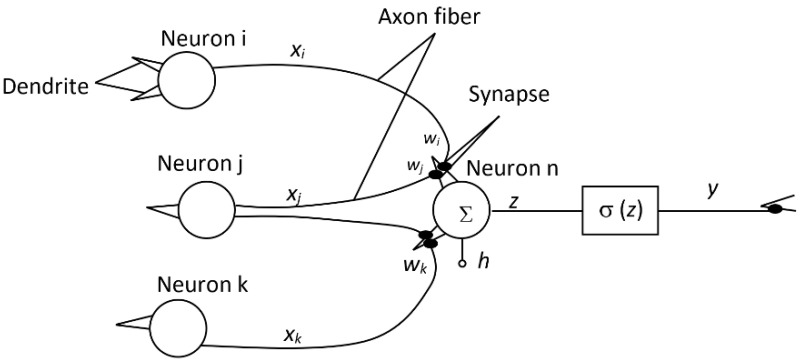
Schema of an artificial neuron inspired by a biological model and how it connects to other neurons.

**Figure 3 materials-15-08234-f003:**
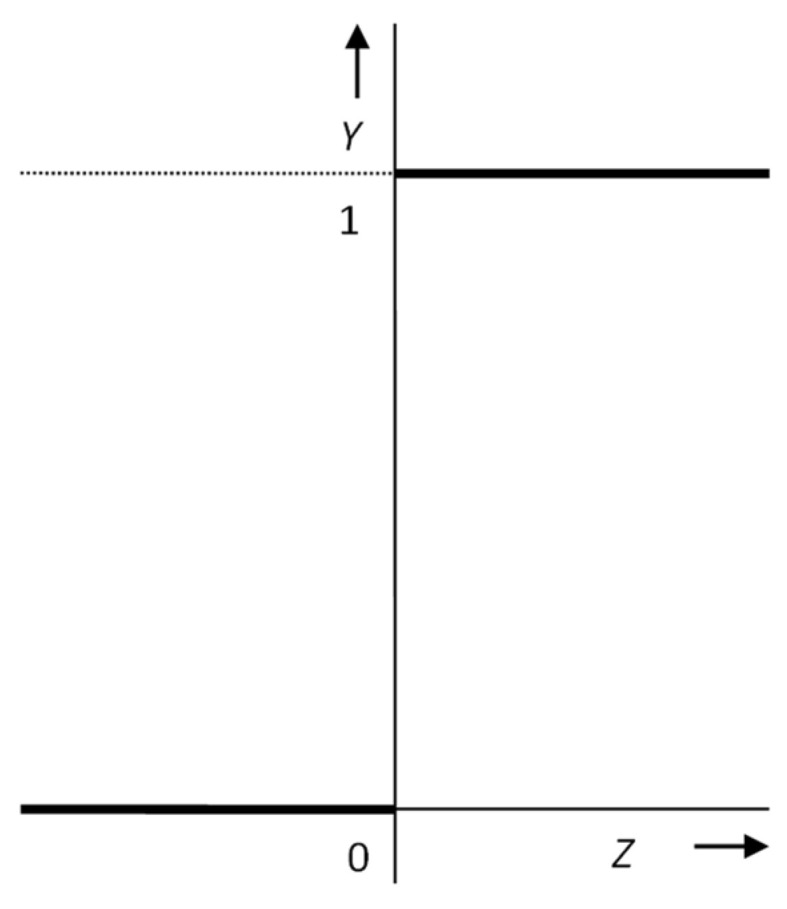
Non-linear Heaviside step function.

**Figure 4 materials-15-08234-f004:**
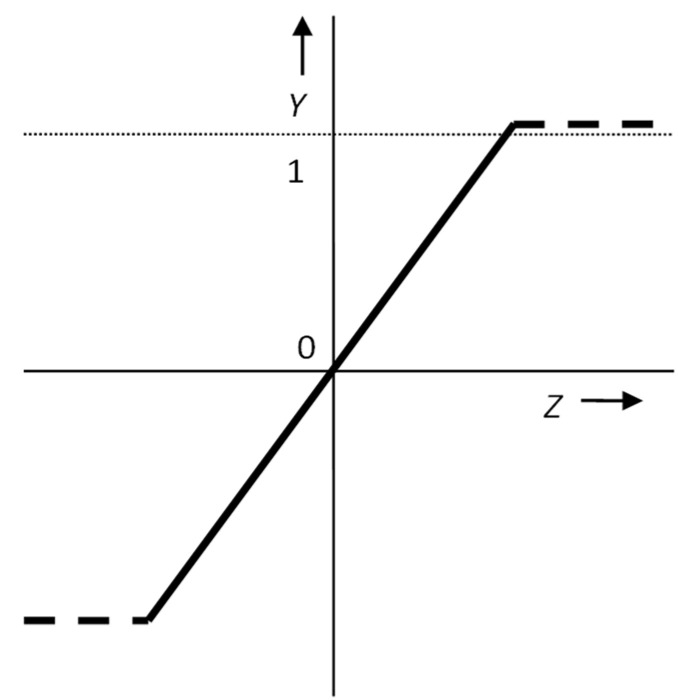
Linear activation function.

**Figure 5 materials-15-08234-f005:**
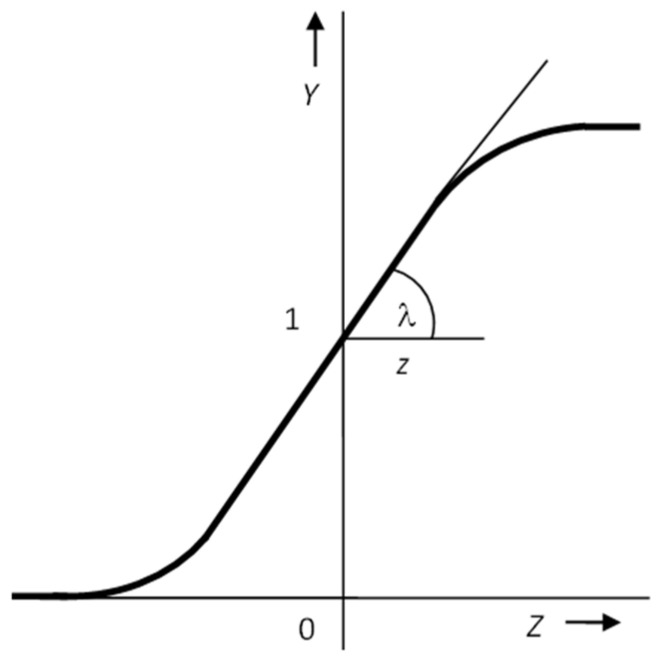
Standard (logistic) sigmoid.

**Figure 6 materials-15-08234-f006:**
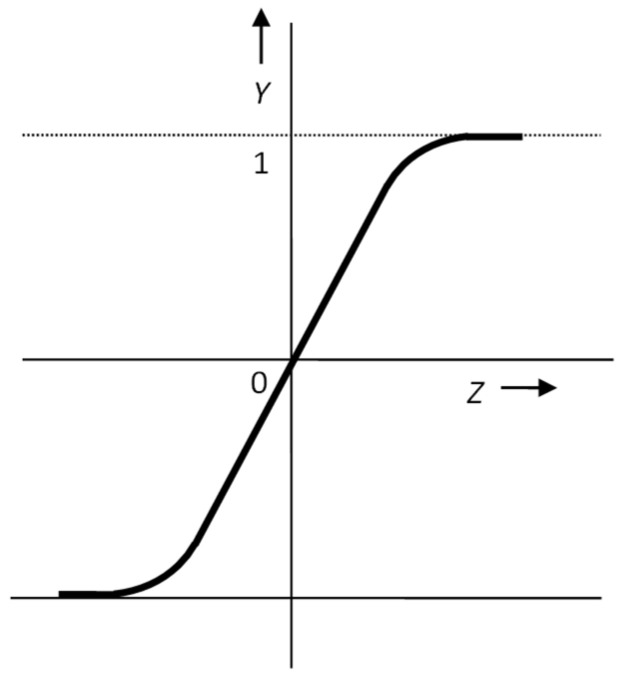
Hyperbolic tangent.

**Figure 8 materials-15-08234-f008:**
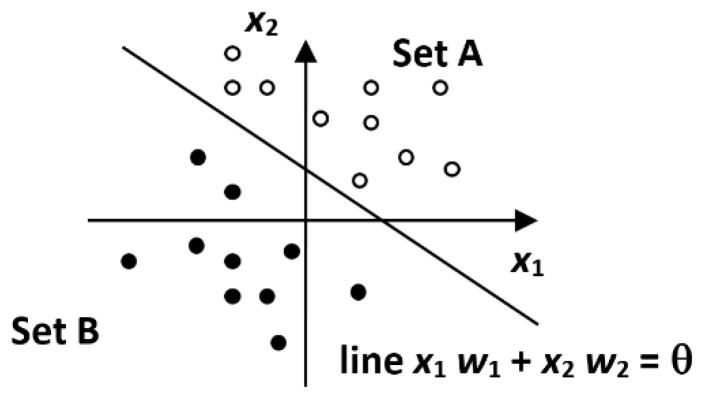
Data classification.

**Figure 9 materials-15-08234-f009:**
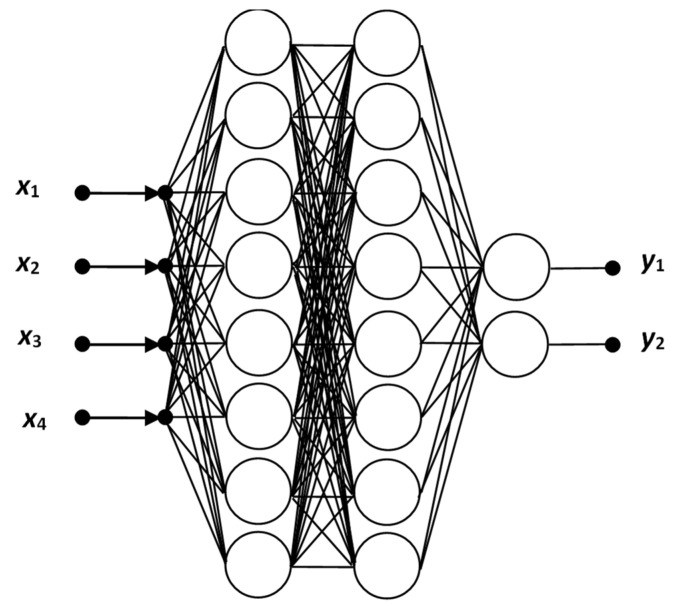
Multi-layer perceptron neural network (MLP).

**Figure 10 materials-15-08234-f010:**
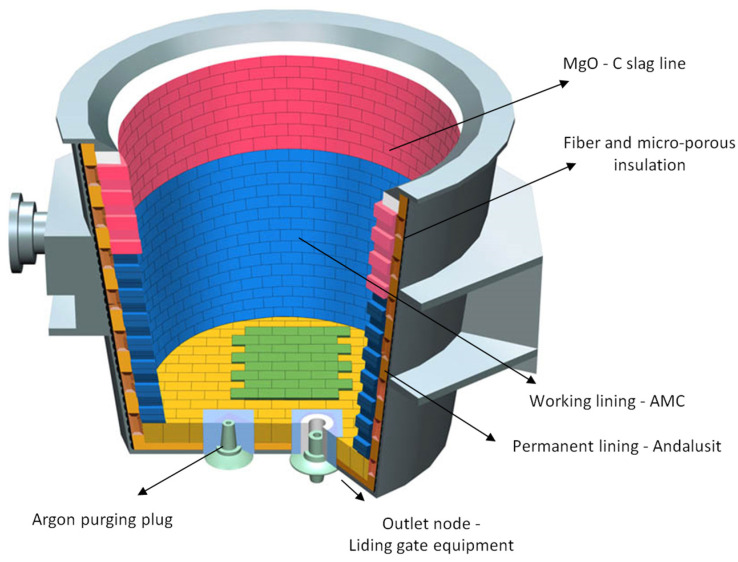
Description of the ladle construction [34].

**Figure 11 materials-15-08234-f011:**
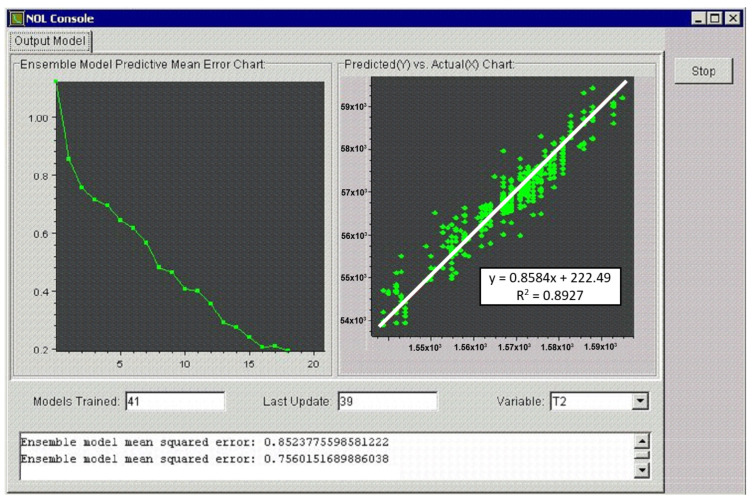
Neural network training progress.

**Figure 12 materials-15-08234-f012:**
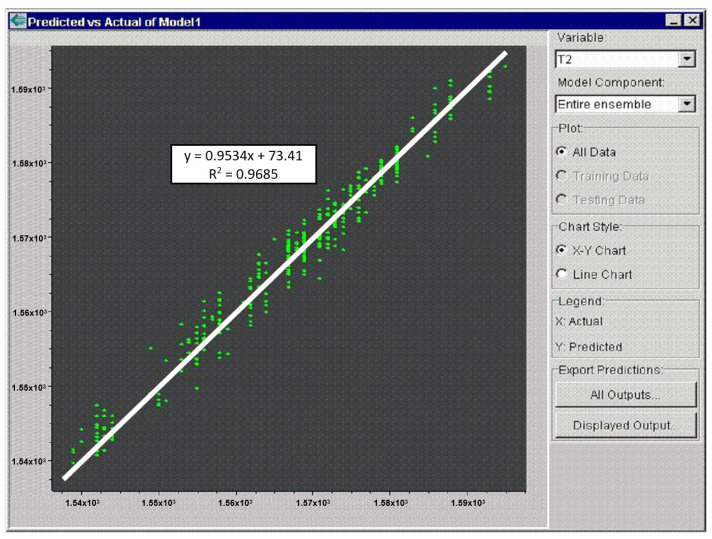
The predicted versus actual value of the model’s output parameter.

**Figure 13 materials-15-08234-f013:**
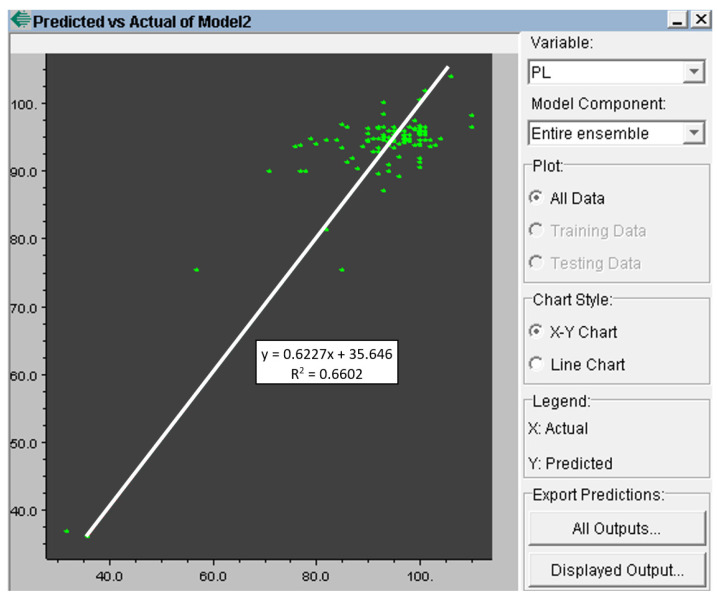
Correlation between calculated (y-axis) and actual (x-axis) ladle lining life.

**Figure 14 materials-15-08234-f014:**
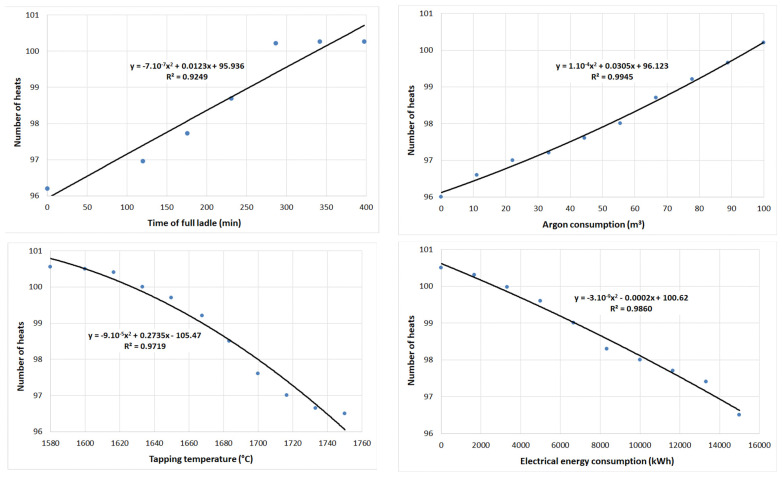
Simulation of the effect of selected operating parameters on lining wear.

**Figure 15 materials-15-08234-f015:**
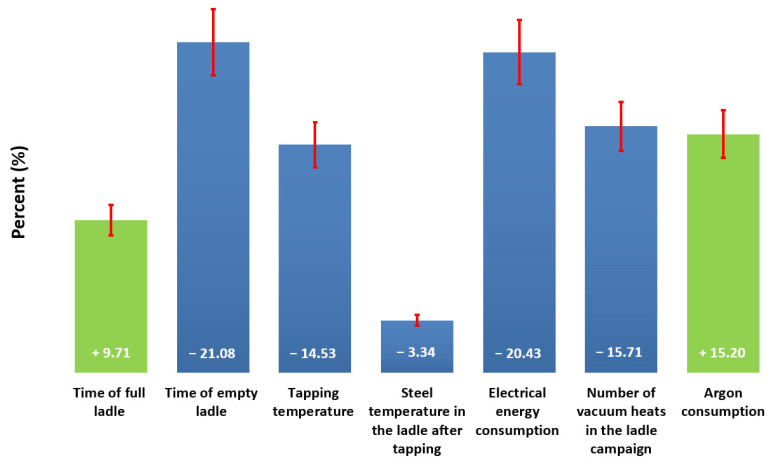
The influence of individual parameters on the life of the lining.

**Table 1 materials-15-08234-t001:** Comparison of the basic properties of a classical computer and a neural network.

Classic Computer	Neural System
the existence of a central processor performing all the work	the central processor has not yet been discovered in nervous tissues and nothing confirms its existence
low parallelization, exceptionally many processors (≤1000) communicating with each other	a very high number of simpler (in terms of information transfer) processors—neurons (≅1012), a very high density of connections between them (min. 10^4^ for each neuron)
exact knowledge of information evaluation and processing (RAM—Random Access Machina, RASP—Random Access Stored Program, Turing machine)	a very vague idea of the activity of individual elements, almost no idea of the way of communication between elements
the necessity of detailed algorithmization of calculations, supported by a deep theoretical background (e.g., the necessity of knowledge of linear algebra for solving systems of linear equations)	it seems that the ability to evaluate different situations is an integral property of nervous systems without the need to understand and be aware of how information is processed
very high speed of elementary operations computing	slow information transfer (on the order of milliseconds)
precisely defined processor architecture	a large number of local elements connected to each other, with great variability in density and method of connection

## Data Availability

The data presented in this study are available on request from the corresponding author.

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
