# Peer review of "Use of Neural Networks for Lifetime Analysis of Teeming Ladles"

_materials, 2022, doi:10.3390/ma15228234_

Round 1

Reviewer 1 Report

The overall concept of the study is good. Please recommend some suggestions for future work at the end of the section "Conclusion".

Author Response

The conclusion was extended with recommendations and suggestions for further research.

Reviewer 2 Report

In this work, the author investigated the lifetime of the teeming ladle by using a neural network. Although a lot of work has been done, the paper has some serious flaws, which are described below in detail. 1. The main innovation is not clear, and from the point of view of writing, the contribution of this paper should be summarized in a paragraph. Please refine the innovation point further. 2. The steps should not be written in sections, and the writing in section 3.1 suffers from such a lack of clarity of expression. Please redescribe Section 3.1 in a clear and simple form. 3. The experiments conducted in this paper are not enough to draw conclusions and put forward views. Please improve the experimental setup or reduce the opinions stated in the conclusion. 4. In the paper, only the positive and negative effects arising from the operational parameters are described, without elaborating the intrinsic causes, which should be further analyzed. Please further analyze the possible reasons for the positive or negative influence of operating parameters on lining wear.

Author Response

  1. The main innovation was specified in the conclusion in a separate paragraph.
  2. Section 3.1 has been simplified and modified.
  3. In the conclusion, opinions of the experiment were limited, based on the results achieved.
  4. At the end of chapter 3, the reasons for the positive or negative influence of operating parameters on lining wear were analyzed.

Reviewer 3 Report

The research does not indicate the composition of the refractory bricks (MgO-C bricks, doloma brick, corundum castable and spinel castable). Slag basicity, high temperature oxidation and erosion are fundamental causes of the different modes of degradation. Studies show that thermal spikes above 1600 OC and acid slags with basicity generally below the ideal of 1.67 cause thermal shock cracking, slag penetration, and phase evolution due to slag-refractory interactions. However, these mechanisms are controlled by diffusion and are not the main responsible for the wear of the bricks, showing that the main responsible for the wear of the bricks is the turbulence of the purging gas, since in the columns adjacent to the gas entry points of purge at the bottom of the ladle up to 86% wear was found in contrast to 24% for the bricks furthest from this effect.

Turbulence from the purging gas is thus responsible for localized and highest lining consumption rate that decimates lining life in these ladles

The network was trained to take the effects of temperature, but it does not take into account the effects of turbulent erosion, which is the main mechanism of brick wear.

Author Response

  1. The specific composition of refractory bricks is detailed in the article.
  2. The authors are aware of the local wear caused by the turbulence of the blown argon at the bottom of the pan. In the research, however, the overall wear of the lining was monitored, on which argon blowing has a positive effect, as shown in the results in Fig. 15. The authors explain the positive effect of argon blowing by thermal and chemical homogenization of the steel, which causes homogenisation of temperatures in the lining of the entire laddle. A specific feature of this steel plant laddles is the impact area  on the wall of the laddle. These areas of the lining suffer the greatest wear from the impacting stream of the steel during tapping from the tandem furnace. The basicity of the slag has the strongest effect on the slag line, which is, for these reasons, made of MgO bricks with 10% carbon. These fittings are known to have the best slag resistance. The wall and bottom of the working lining is made of AMC fittings, which have proven themselve the most in the steelworks, due to the large downtimes of the casting ladle.

Reviewer 4 Report

The submission is quite interesting, it was a pleasure to read it.

I have a question concerning figures 11, 12 and 13. Maybe instead of showing screenshots from the software, it would be better to present the date on separate graphs? It would be much clearer. Right now the red regression line is almost invisible. Additionally, please add the formula of the regression line and the correlation coefficient.

Concerning figure 13 was the correlation coefficient R or R2?

 In figure 14 the data points should not be connected. You can add a regression as well if you want.

In figure 15 please add error margin bars.

Author Response

  1. Viewing the image from the software of NS Studio gives us a better understanding of the method and a more accurate characterization of the software used. We changed the red curve to white for better visibility and added the formula of the regression line and the correlation coefficient.
  2. The correlation coefficient is R2, we have added this information to the text.
  3. The figure 14 was modified according to the reviewer's instructions and the regression dependencies were added.
  4. In Figure 15, error margin bars were added according to the reviewer's instructions.

Round 2

Reviewer 2 Report

1.     In the article, each table needs a table header to explain the content of the table, please add a table header to the table.

2.     Please use serial numbers instead of dots to mark the serial numbers of the subsections, for example, this problem occurs in sub-section 3.1 and is described in the text below Figure 10.

3.     Please check and correct the insertion of references, some of them are wrong.

4.     English expression needs to be improved.

Author Response

  1. The header of table 1 was added to the article.
  2. The authors prepared their manuscript following the Template journal Materials. It allows writing in two ways, a bulleted list and a numbered list. After consultation, the authors decided to prefer a bulleted list option for better clarity.
  3. Erroneously inserted references have been corrected in the text.
  4. The authors checked the English expression and adjusted and clarified some terms.

Reviewer 3 Report

The authors explain the positive effect of argon blowing by thermal and chemical homogenization of the steel, which causes homogenization of temperatures in the lining of the entire laddle. This confirms that the network was trained only to take the effects of temperature variations and that all the variables analyzed are reflected in temperature variations. 

During an argon agitation, the injection is carried out mainly in two phases, a first one that is a strong flow to homogenize and a second phase with a lower flow but a longer time for cleaning the steel, in addition there is the need to make adjustments, because after several castings the wear of the porous plugs affects the agitation of the steel. Therefore, I would recommend the authors to define in their work that the agitation with the argon injection is handled as a constant.

Author Response

The authors are very grateful for the valuable advice and comments that will be used in further research. In the text, the sentence is added that agitation with the argon injection is handled as a constant.